# The Efficiency of UV Picosecond Laser Processing in the Shaping of Surface Structures on Elastomers

**DOI:** 10.3390/polym12092041

**Published:** 2020-09-08

**Authors:** Bogdan Antoszewski, Szymon Tofil, Krystian Mulczyk

**Affiliations:** Kielce University of Technology, Department of Mechatronics and Machine Building, Centre for Laser Technologies of Metals, al. Tysiąclecia Państwa Polskiego 7, 25–314 Kielce, Poland; ktrba@tu.kielce.pl (B.A.); kmulczyk@tu.kielce.pl (K.M.)

**Keywords:** UV laser, picosecond laser, elastomers, micromachining

## Abstract

Elastomers are used as construction materials in numerous industries, and in particular the biomedical industry, mechatronics, electronics, the automotive industry, and chemical devices. The paper presents the results of tests involving the effects of microprocessing of elastomeric materials using a UV laser emitting picosecond pulses. In particular, it presents an analysis of the influence of the parameters of processing on its efficiency. The paper provides a recommendation of the most advantageous processing parameters for materials such as polyurethane and silicone (MVQ). The authors see prospects for the use of the developed technology in the techniques of sealing and microfluidisation. The final part of the paper presents examples of surface structures generated on elements made of artificial materials and the results of tests involving reduction of friction resistance of sealing rings in a pneumatic actuator.

## 1. Introduction

The general trend of miniaturisation also involves elements made of elastomers. The wide use of elastomeric materials is due to the specific properties of these materials, such as deformability, elasticity, chemical durability, resistance to atmospheric conditions, good electric insulation, and sometimes biocompatibility and resistance to heat. They are used in biomedical, mechatronic, chemical, and electronic industries, where the manufactured elements include, e.g., microflow systems, surfaces with specific adhesive properties, chemical and biological microreactors. Practically, it should be concluded that mechanical methods for shaping the surface structures of elastomers in this case are inefficient.

In such a situation, absorbency is effectively improved by colouring the material with pigments. The issue of absorbency loses its significance when using lasers emitting UV radiation or excimer lasers. In scientific literature, there are an increasing number of papers related to laser processing of elastomeric and generally polymeric materials [1,2,3,4]. Laser microprocessing of elastomeric materials [5,6,7,8,9,10,11,12,13,14,15] is an alternative to lithographic methods; it belongs to direct methods, using the sources of laser radiation, usually: CO_2_ (wavelength 10.6 μm—medium infrared) and Nd-YAG (consecutive wave harmonics: 1064, 532, 326, and 266 nm—from near infrared to ultraviolet), also including ultrashort pulse lasers.

## 2. Theoretical Bases

The direct impact of a laser spot on the surface of a material results in, depending on the type of the material and the parameters of the laser beam (wavelength, pulse duration, repetition frequency, etc.), burning, melting, sublimation, depolymerisation (PMMA), ablation [16,17,18,19,20,21,22,23,24,25,26,27] (in particular desorption), foaming, colour change, and annealing (metals). The traditional method of direct processing with a CO_2_ laser beam has already been used for several years. Unfortunately, it has serious disadvantages: low quality of the resulting structure—high roughness of the processed surface, heavily dependent on the material used, and a minimum spot diameter of 100 μm, which limits the field of application considerably. The improved TC&T method (Laser Through-Cutting and Pattern Transfer) for direct CO_2_ laser processing was developed by Liu and Gong [28]. Its application enabled the generation of microchannels with a minimum diameter of 30 μm, with smooth surfaces of their bottoms and sidewalls.

Laser-based methods for processing the surfaces of polymeric materials [1,2,29,30,31,32,33,34,35,36,37], using radiation to generate heat effects, involve the melting of surface microareas, several micrometres in thickness, increasing roughness of the surface and removing its fragments, and if the process takes place in the presence of oxygen, surface oxidation also occurs. This method works for polyamides, polyesters and aramid materials, or for polyolefins and PTFE, if ultraviolet absorbers are introduced into them, since the coefficient of absorption of laser radiation by the surface of a polymeric material decides about the effectiveness of this technology.

Laser radiation is used for very precise processing of small surfaces with complex geometrical forms, e.g., in electronics, or in the manufacturing of medical devices, where it additionally ensures their sterilisation. Laser radiation influences an improvement in wettability and the value of surface free energy by the implementation of polar function groups, primarily in the oxidation process, the degree of polymer crosslinking and the type of geometrical structure of the surface, with no change in properties of the core.

When processing polycarbonates, polyethylene terephthalate or polystyrene, it is required to use a laser radiation wavelength of less than 308 nm, which is provided by an ArF excimer laser (GAM Laser, Orlando, FL, USA). When using radiation energy which is lower or higher than the ablation threshold of the listed polymeric materials, respectively, an improvement in the hydrophilic or hydrophobic properties is ensured, which is accompanied by an increase in roughness [2,3,4,5,26].

Laser ablation constitutes a process which results in severing the chemical bonds of polymeric material macroparticles due to the impact of laser light. There are two distinguishable ablation mechanisms which can occur separately or simultaneously: photochemical and photothermal. The process of thermal ablation of a polymer occurs as a result of the excitation of particles to a high-energy state by laser radiation. This results in the generation, and in the next step accumulation, of heat and an increase in the temperature of the material. The process of thermal ablation occurs after exceeding a material temperature value called the threshold ablation temperature (*T*_D_). Energy initiating the process of thermal ablation is expressed by the following equation:(1)$$E_{j}^{th}=c_{\nu }\frac{\left(T_{D}-T_{R}\right)}{\alpha (1-R)}$$
where: *T*_R_—initial temperature of polymeric material, *c_v_*—specific heat capacity of polymeric material, α—radiation absorption coefficient, and *R*—laser radiation reflection coefficient.

The mechanism of photochemical ablation involves photolytic fracturing of chemical bonds, especially C–H bonds. Ablation reaches a full extent of its development when, as a result of a laser pulse, a large number (*n*) of chemical bonds crack at the same time. The value of energy initiating ablation (threshold value of ablation energy) is described by the equation:(2)$$E_{j}^{th}=n\frac{h\nu }{\varphi \alpha (1-R)}$$
where: *ϕ*—quantum yield of bond scission (value from within a range of 0–1) and *hν*—photon energy. 

## 3. Methodology

The research presented in the paper proceeds according to the following pattern. Samples of tested materials are subjected to laser processing according to the plan of the experiment. The plan of the experiment assumed the generation of a series of indentations with circular cross-sections, with a constant diameter and depth, with variable operating parameters of the laser (P—laser power, v—scanning speed, f—pulse frequency). A single indentation was generated by scanning four coaxial circles with decreasing diameters by means of a laser beam (Figure 1A). During the movement of the beam, L_n_ consecutive laser pulses were hitting the spot footprint of a single laser pulse (Figure 1B).

Upon removing the products of processing in an ultrasonic cleaner, the samples were subjected to microscopic examinations. During processing, we deal with the overlapping of individual pulses with a Gaussian energy density distribution for one beam trajectory (a single cavity). The mutual overlapping of the pulses depends on the operating parameters of the laser device and increases with the increase of the pulse energy. The efficiency of the process also depends on the diffraction of radiation on the walls of the cavity, as well as the interaction of radiation pulses with the material particles detached from the treated surface. Geometrical parameters of the results of processing (diameter, depth, removed volume) were identified by means of a HIROX KH-8700 microscope (Hirox Co., Ltd., Tokyo, Japan). A qualitative assessment of the resulting microstructures was performed based on observations using a Joel electron microscope. Calculations of the processing parameters were performed according to relationships presented in Table 1.

where in:*v* —speed (mm/sec)*P*—laser power setting (W)*f*—pulse frequency (kHZ)*Φ_L_*—spot diameter (µm)—here assumed as 40 (µm)*A_L_*—spot surface area (µm^2^)*Q*—volume of removed material (mm^3^)*s*—processing path length for one microindentation, equal to 0.6912 mm

Calculations according to the formulae presented above were used for a detailed processing efficiency analysis. Estimation of the geometrical parameters of processing results was accomplished using a HIROX digital microscope along with its software, and a JOEL electron microscope (JEOL Ltd., Tokyo, Japan).

Polyurethane (PU) and silicone (MVQ) were selected for the tests as materials with a considerable application potential. This was based especially on the application of these materials as seals in pneumatic and hydraulic devices.

### The Experiment

The experimental tests were performed in a testing station existing in the Centre for Laser Technologies of Metals of the Kielce University of Technology. The station constituting a laser processing machine for microprocessing is presented as a layout in Figure 2. Figure 2 shows the changes in the view of the microindentations with changes in frequency and pulse energy. Visible here are microindentations with various dominant ablation mechanisms.

The characteristics of primary components of a laser processing machine for microprocessing are as follows:

TruMicro 5235c Laser
-laser type: a diode-pumped pulse disk laser with third-harmonic generation,-wavelength: 343 nm,-average power: 5 W,-minimum pulse duration: 6.2 ps,-pulse frequency: 400 kHz, with a possibility to divide by a natural number from 1 to 10,000,-maximum pulse energy: 12.6 μJ,-mode: TM_00_,-M^2^ = 1.3, and-maximum fluence: 4.8 J/cm^2^.

IntelliSCAN 14 scanner
-biaxial scanner (XY) for ultraviolet radiation of 343 nm,-aperture: 14 mm,-a standard F-Theta lens with a focal length of: f = 160 mm,-Rayleigh length—2.76 mm, and-spot diameter—18.2 µm.

Two materials used in the construction of seals selected for the tests were characterised by their considerably differing properties. Silicone MVQ and polyurethane PU are materials frequently used as seals, especially in the chemical industry (Table 2).

The objective of the performed tests involved determining the efficiency of surface microprocessing of selected elastomeric materials in terms of shaping structures which are useful in the technique of sealing.

A test involved the generation of microindentations on the studied materials with the energy of laser pulses ranging from 3.75 to 12.5 µJ and frequencies of 400 and 200 kHz, as well as with scanning speeds varying from 10 to 640 mm/s. The processing was carried out in the form of concentric circles with radii increasing in sequence by a value equal to the focal diameter of the beam, amounting to 30 µm. The objective of the processing involved the generation of indentations with diameters of approximately 200 µm, with their bottoms as flat as possible. The effects of processing were identified by means of analyses performed under a HIROX microscope (Hirox Co., Ltd., Tokyo, Japan). A sample view of an indentation observed via a HIROX microscope and the results of its geometrical parameters are presented in Figure 3. The HIROX microscope software enabled the determination of the volume of the resulting microindentations. Figure 4 presents a sample view of indentations for PU and MVQ observed under a JOEL microscope.

## 4. Analysis of the Results

The analysis was performed based on the results of the experiment and photomicrographs, plots and calculations performed according to relationships presented in Table 1.

The investigated case of elastomeric materials deals with various specific ablation mechanisms. Apart from the primary mechanism involving the evaporation of material, there is also thermal degradation and degradation caused by the impact of high-energy radiation. Both of these multi-stage mechanisms involve the scission of primary chains, depolymerisation, the generation of volatile products and coke. As seen in Figure 4, destructive processes for both tested materials present considerably different images. For the PU material, thermal degradation is of prevailing significance, while in the case of MVQ, a major function is served by the crosslinking processes of the evaporation of volatile components. Processing products in the form of coke are generated in both cases. This is evidenced by weak processing results with low energy parameters of the laser radiation beam, as well as the remains of material particles on the bottom and walls of an indentation. The course of the relationship between the removed volume and the linear density of laser radiation indicates that the nature of this relationship is asymptotic up to a certain maximum value of the removed volume. This means that, for high values of linear energy density of laser radiation (which corresponds to low scanning speeds), a further increase in linear energy density results in no significant increase in the removed volume (Figure 5). Figure 5 shows the linear energy, not the scanning speed, because in fact the value of the material loss depends on the value of the energy involved in the movement of the beam. This approach covers all machining parameters. The relationship between the removed volume and the number of pulses is of a similar nature. Considering the impact of pulse frequency on the efficiency of processing (Figure 6), it should be noticed that the removed volume will go up along with an increase in frequency. This is affected by the energy accumulation phenomenon and it is related to the heating of the processed material as a result of absorbing the energy of laser pulses. This results in lowering the effective threshold ablation energy density, and as a consequence, in increasing the efficiency of the process. Therefore, reduction of the time between subsequent pulses due to lower heat dissipation will result in increased efficiency of the process.

The reduction in processing efficiency for PU observed in Figure 6 is due to two factors—the diffraction of the laser beam at the edges of the hole with a high slenderness ratio and the interaction of the removed material particles with successive pulses incident on the surface of the processed material.

A similar phenomenon would occur for the MVQ material with the use of higher processing parameters impossible to achieve at the available position.

Figure 7 showing the effect of treatment effects depending on the number of pulses of a given energy has a similar course as the relationship shown in Figure 5, which indicates a similar effect of changing the linear laser energy and the number of pulses.

The produced results indicate that a laser emitting UV radiation in picosecond pulses is an efficient tool for microprocessing of elastomeric materials: PU and MVQ. The values of threshold ablation energy estimated based on the results of the experiment and calculations amount to 1.9 to 2.2 µJ for the MVQ material and 1.6 to 1.8 µJ for the PU material, respectively. For these energy ranges a frequency of 400 kHz and a scan speed of 160 mm/s were used.

PU is the material which is more susceptible to processing, since achieving a 2 min·µm^3^ reduction with the same scanning speeds requires energy in the order of 5.2 µJ, while MVQ needs as much as 9 µJ. It should be noted that the efficiency of processing is affected by the colour of materials, and therefore by varying absorption of radiation. The quality of processing was assessed based on an analysis of changes in the profile of the bottom of a microindentation. The disappearance of concentric circles generating the waviness of the bottom of a microindentation was assumed as a criterion of achieving acceptable quality. The processing parameters which enabled the achievement of acceptable quality and maximum efficiency were chosen as the recommended ones. The device at our disposal generates a maximum energy of 12.5 µJ and a maximum pulse frequency of 400 kHz and we are not able to perform tests above this value. It should be assumed that the efficiency of processing with increasing these parameters increases—this applies in particular to MVQ material. The possible increase in machining efficiency depends on the slenderness ratio of the pit. They amounted to v = 80 mm/s, E_i_ = 12.5 µJ for the PU material and v = 160 mm/s, E_i_ = 12.5 µJ for MVQ, respectively, for a frequency of 200 kHz, and 160 mm/s, 12.5 µJ for both materials for a frequency of 400 kHz.

## 5. Example of Application

Constant optimisation of production processes causes a continuous increase in requirements faced by elastomeric seals. These requirements can be highly diverse and they depend not just on the type of application, but also on the industry. In general, the usefulness of a sealing material for a given application is decided by three groups of factors: operating temperature, chemical durability and mechanical properties. One should also consider mutual impacts, e.g., that of operating temperature and mechanical properties. Apart from proper selection of material, the effectiveness of a seal may also be decided by issues such as, e.g., design, geometry of housing, as well as the physical, chemical, and stereoscopic properties of the surface. This primarily concerns those seal surface fragments which determine the tightness of the contact between the seal and the surface of a machine element or the housing. They are mainly small fragments of the seal, such as various shapes of sealing flanges, as well as cylindrical and flat surfaces. The shaping of surface properties of these seal fragments, including those related to adhesion, is of primary significance for the course of processes in a seal crack. It is possible with the use of laser microprocessing. In many cases, for seals of rotary shafts (PUWO) made of harder materials, e.g., PTFE, in order to prevent fast cutting of a groove under the flange, it is necessary to use shafts hardened to 58–62 HRC (Hardness Rockwell C). For dry seals, it is recommended to grease the seals during assembly with a proper lubricant or use air lubricators. Due to the lack of lubrication, the seal may become jammed (stuck) in the housing. These problems can be largely solved by the use of elastomeric material microprocessing in the place of abutment of a seal fragment and the surface of a machine element or housing. The shape of microprocessing may have a varying nature. These can be lubricant reservoirs shaped as domes, grooved profiles shaping microscopic blades, or sets of flow-dosing channels. These types of solutions can be useful, e.g., when eliminating the stick–slip phenomenon, and the prospect of using microprocessing combined with flexibility and dynamics of sealing seems to be very promising.

Considering the above, an attempt was made in the paper to test the effectiveness of reaction of surface texture to friction resistance of the seal in a pneumatic actuator piston. Three rows of peripheral microindentations were generated on the flange of a seal made of the MVQ material, abutting the cylinder. Figure 8 presents the positioning of microindentations and their geometrical parameters, and Figure 9 presents the real view of the resulting texture (operating parameters of the laser device are: pulse energy 12.5 µJ and scanning speed 160 mm/s). The distance between the edges of adjacent microindentations amounted to approximately 80 µm.

The efficiency of the generated texture under the conditions of friction was tested experimentally by assessing the idling resistances of a pneumatic actuator piston. Using the INSTRON (Norwood, MA, USA) strength machine an experiment was performed to test actuator movement resistance with a textureless and textured seal. The surfaces of previously defatted piston seals were covered by a single layer of oil mist, and the recording of movement resistance was initiated after a 5-min operation of the actuator. The recorded results are presented on a plot in Figure 10.

As expected, higher friction force values were observed in turning points. Average friction force values for pistons with textured seals were lower than those of pistons with textureless seals by a value of 22% (Figure 10). This effect is due to the improvement of lubricating conditions. The microindentations constituted lubricant reservoirs, and at the same time they were the source of advantageous hydrodynamic effects. It should be noted that the experiment was performed during idling of the engine, which means no load of force or pressure. If higher pressures against cylinder walls occur, the impact of texture on friction resistance will be even more noticeable. The presented solution may be used primarily in precise actuators of control systems, where resistance is of high significance, and in any place where the stick–slip phenomenon occurs. The research on the influence of the texture of the working surface of the mechanical seals shows that the texture does not deteriorate the durability of the seal. The rationale is as follows;
-texture elements create reservoirs of the lubricant and provide better lubrication,-debris and wear particles can accumulate in the texture cavities and are thus eliminated from the friction zone, and-texture depressions under favorable circumstances are the source of lubrication-improving hydrodynamic effects.

Test results for the durability of mechanical seals with a textured working surface show increased durability. The authors believe that the same may be the case with elastomeric seals. Detailed identification of this problem is planned in further research.

## 6. Summary

The performed tests and observations proved that a laser emitting UV radiation within a range of 343 nm in picosecond pulses can be recommended as a tool for microprocessing of elastomeric materials, and in particular PU and MVQ. The microprocessing of indentations is efficient and precise, and the processed material does not lose its elastic properties. No charring or other traces indicating the overheating of the material were noticed in the place of processing and its surroundings.

The estimated value of threshold ablation energy for the MVQ material ranges from 1.9 to 2.2 µJ and from 1.6 to 1.8 µJ for the PU material. The recommended processing parameters amount to v = 80 mm/s, E_i_ = 12.5 µJ for the PU material and v = 160 mm/s, E_i_ = 12.5 µJ for MVQ, respectively, for a frequency of 200 kHz, and 160 mm/s, 12.5 µJ for both materials for a frequency of 400 kHz.

Application of the texture described above on the rings of a piston in a pneumatic actuator results in a 22% reduction in friction resistance.

## Figures and Tables

**Figure 1 polymers-12-02041-f001:**
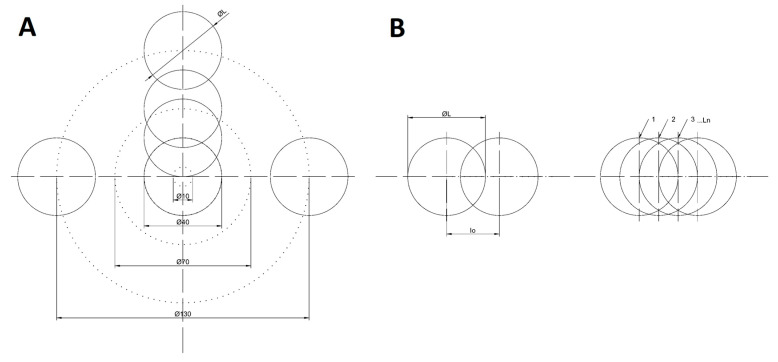
(**A**) Laser beam trajectory during the generation of a single indentation, (**B**) the overlapping of single pulses. ØL—beam spot diameter, I_o_—pulses overlap

**Figure 2 polymers-12-02041-f002:**
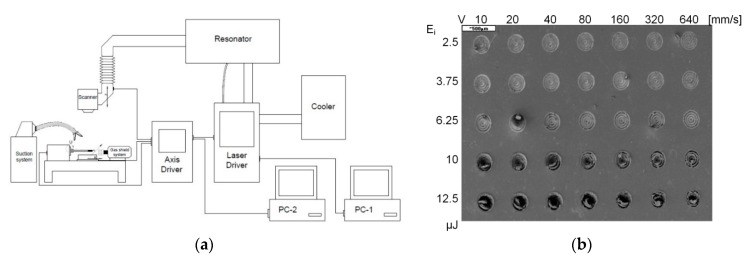
The layout of a testing station with a TruMicro5325c laser (**a**) and a general view of a sample with a generated matrix of microindentations (**b**).

**Figure 3 polymers-12-02041-f003:**
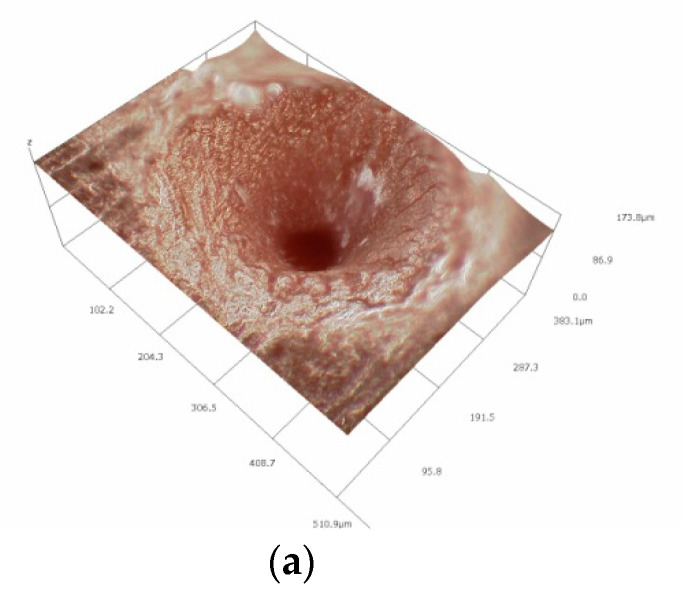
The appearance (**a**) and geometrical parameters (**b**) of a sample indentation from observations under a HIROX KH-8700 microscope.

**Figure 4 polymers-12-02041-f004:**
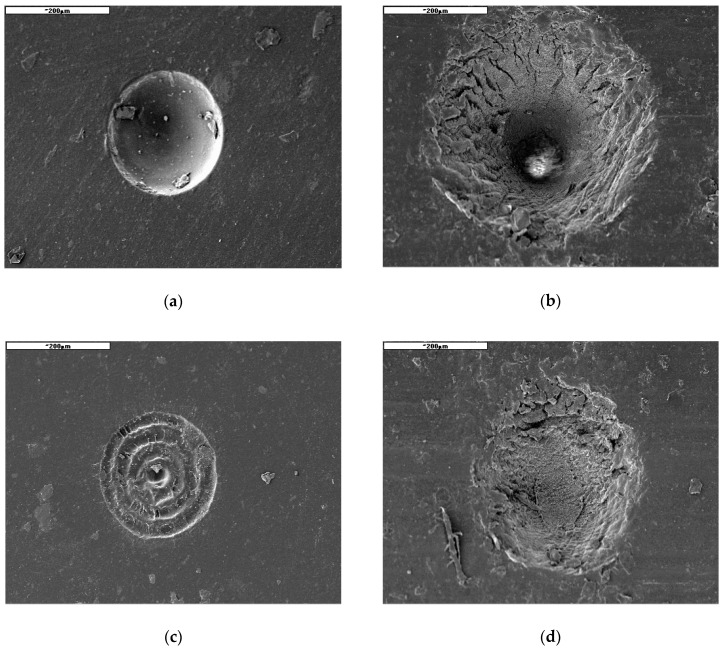
Views of sample indentations generated by means of the SEM method on gold-sprayed samples in: (**a**) Polyurethane (PU) with high energy values, (**b**) silicone (MVQ) with high energy values, (**c**) PU with low energy values, (**d**) MVQ with low energy values, (**e**,**f**) expelled material particles visible.

**Figure 5 polymers-12-02041-f005:**
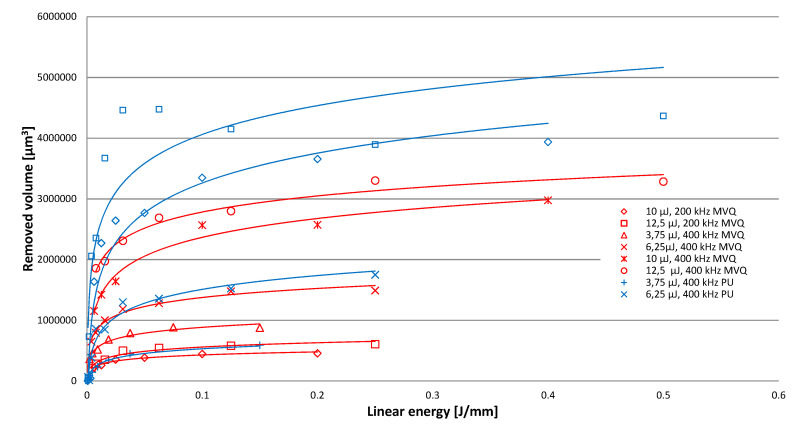
Relationship between the volume of microindentations and linear energy density of laser radiation for the PU and MVQ material.

**Figure 6 polymers-12-02041-f006:**
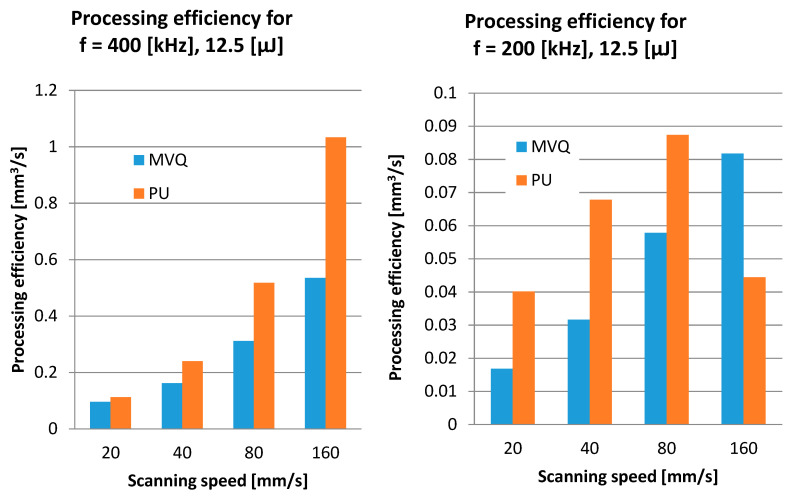
Processing efficiency for PU and MVQ (Q/t = f (200 and 400 kHz) for four selected scanning speeds.

**Figure 7 polymers-12-02041-f007:**
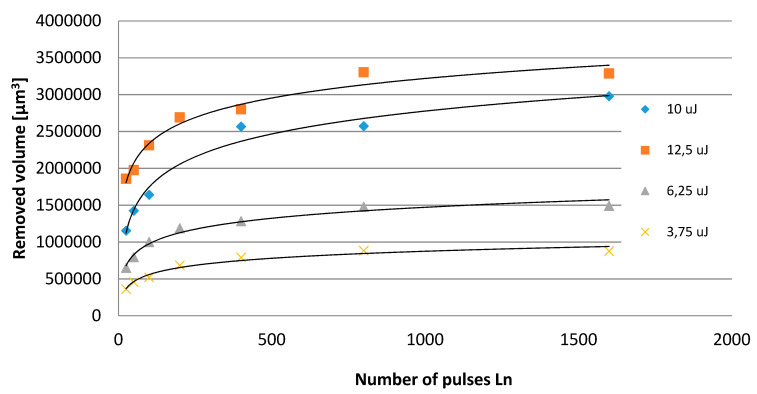
The volume of a single indentation depending on the number of pulses with various scanning speeds of MVQ material.

**Figure 8 polymers-12-02041-f008:**
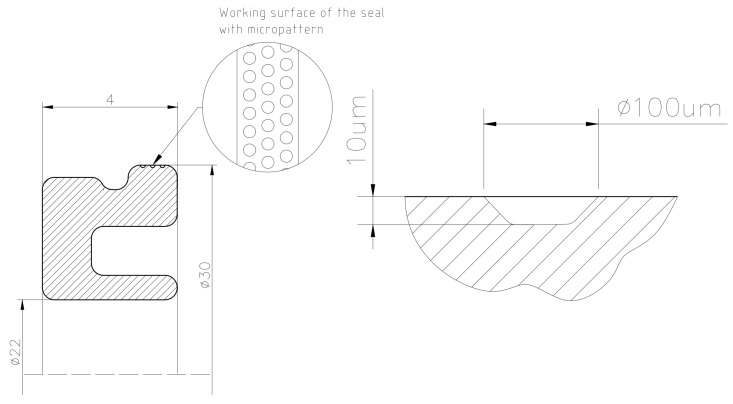
A pattern of the distribution of microindentations on the sealing ring of a piston, (**left**) a cross-section of the sealing ring with indication of the place of processing and (**right**) dimensions of a single microindentation.

**Figure 9 polymers-12-02041-f009:**
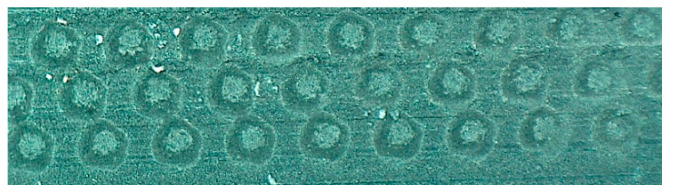
Real view of texture on the outer surface of the ring.

**Figure 10 polymers-12-02041-f010:**
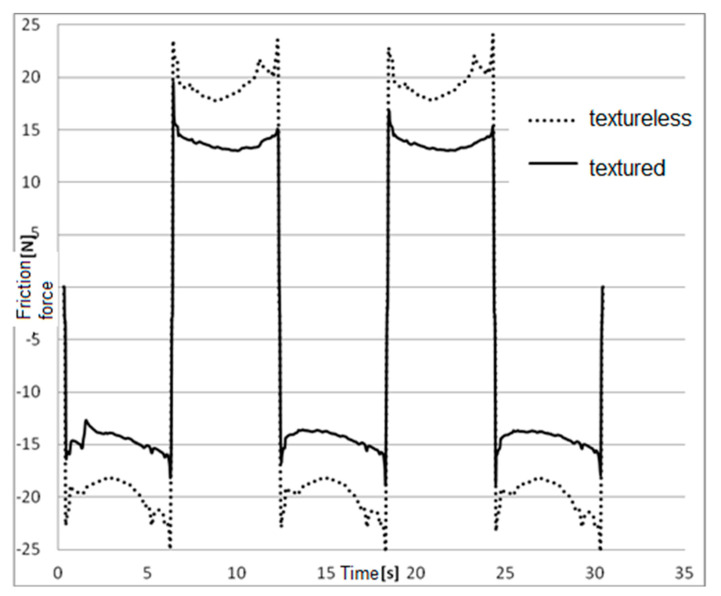
Friction force values for a ring without texture (dashed line) and with texture (solid line).

**Table 1 polymers-12-02041-t001:** List of relationships used in the analysis.

$P_{śr}=\frac{P}{pulse\;divisor}$	*P_śr_* —Average Power (W)
$E_{e}=\frac{P_{śr}}{f}1000$	*E_e_* — single pulse energy (µJ)
$E_{l}=\frac{P_{śr}}{v}$	*E_l_*—linear energy density of laser radiation (J/m)
$I_{o}=\frac{v}{f}$	*I_o_*—distance between pulse centres (µm)
$L_{n}=\frac{\emptyset_{L}}{I_{o}}$	*L_n_*—number of pulses per spot
$F=\frac{I_{e}\left(\emptyset_{L}-I_{o}\right)}{A_{L}\emptyset_{L}}100$	*F*—fluence (J/cm^2^)
*t* = *s*/*v*	*t*—processing time of a single indentation (s)
*W* = *Q*/*t*	*W*—processing efficiency (µm^3^/s)
$N_{imp}=f*t$	*N_imp_*—number of pulses per a single indentation

**Table 2 polymers-12-02041-t002:** Tested materials—selected properties.

Name of Material	Density of Material (kg/cm^3^)	Hardness (Shore Scale)	Temperature of Use (°C)	Colour of Material
Polyurethane PU	1.20	80	−30 ÷ +80	yellow
Silicone MVQ	1.28	60	−40 ÷ +240	red

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
