# Peer review of "The Efficiency of UV Picosecond Laser Processing in the Shaping of Surface Structures on Elastomers"

_polymers, 2020, doi:10.3390/polym12092041_

Round 1
Reviewer 1 Report
The presented paper is aimed at studying the UV laser ablation process for some elastomeric material. The subject of the work is very specific, and therefore the paper could be considered a very good technical report rather than a scientific paper. However, the paper has some merit, which could allow its publication on the journal.
There are different issues which should be solved before suggesting publication:
The basics for the development of eq. 1 and 2 should be given, otherwise please provide proper reference
Row 88 what are P, v and f? they are defined later, but they should be defined at this stage
Table I: most of the reported parameters are not cited in the text of the paper or in the figures. Probably some of the reported relations could be removed?
Figure 3 label as a and b. it is clear that the profile of indentation is reported in b), however, the unit measures should be given
Table 3 the number of significant digits is very high. What do the percentages represent?
Figure 2 b), rather than the view of a sample with indentations, probably it would be better to report the scanning patterns for microindentation generation
Table 3 and 4 are not necessary, since the results presented are replicated in figure 5. I don’t understand why, in Figure 5, the linear energy is reported and not the scanning speed. Figure 5 should have its corresponding for PU
Figure 7 is not cited in the text. also, figure 7 presents the results relative to which material?
Figure 6: it is very strange the result for PU at 160 mm/s which shows a reduction of the processing efficiency, and is not discussed by the authors.
In general, tests should be performed at different replications, and data should be provided with the corresponding standard deviations
Row 207-208 same scanning speed? Which speed? And which frequency?
Row 214-216. The problem here is that the optimal chosen parameters correspond to the limits of the operating conditions. For example, at 400 kHz a scanning speed of 160 mm/s is suggested. However, it is not clear what happens if the scanning speed is increased. Analogously, an energy of 12 mJ is suggested, but what happens if the energy is increased? According to the results of figure 6 the processing efficiency is expected to increase.
Figure 8 a) is not clear, in particular it is not evident the position of microindentation. The overall dimensions of the sealing ring should be given
Figure 10 is not necessary, probably it would be better to report a scheme of the test better highlighting the piston and seal
Figure 11 shows a reduction of the friction force by the use of indented seals. However, it is not clear what would be the effect on the durability of the seals. The presence of indentations on the surface of the seal is expected to reduce the service life, do to easier material ablation during operation. This issue should be better addressed and the potential advantages of the developed approach should be compared with potential issues
Author Response
Responses to the review.
- The basics for the development of eq. 1 and 2 should be given, otherwise please provide proper reference.
The source of the equation were from references number 2. The reference was completed
- Row 88 what are P, v and f? they are defined later, but they should be defined at this stage.
The listed values have been explained.
- Table I: most of the reported parameters are not cited in the text of the paper or in the figures. Probably some of the reported relations could be removed?
The given values are used in the paper.
- Figure 3 label as a and b. it is clear that the profile of indentation is reported in b), however, the unit measures should be given.
Drawing caption has been changed.
- Table 3 the number of significant digits is very high. What do the percentages represent?
The table has been deleted.
- Figure 2 b), rather than the view of a sample with indentations, probably it would be better to report the scanning patterns for microindentation generation.
Drawing caption has been changed.
- Table 3 and 4 are not necessary, since the results presented are replicated in figure 5. I don’t understand why, in Figure 5, the linear energy is reported and not the scanning speed. Figure 5 should have its corresponding for PU.
The table has been deleted. The results for PU are shown in blue in the graph. Chart 5 shows the linear energy, not the scanning speed, because in fact the value of the material loss depends on the value of the energy involved in the movement of the beam. This approach covers all machining parameters.
- Figure 7 is not cited in the text. also, figure 7 presents the results relative to which material?
The description has been completed. Fig. 7 showing the effect of treatment effects depending on the number of pulses of a given energy has a similar course as the relationship shown in Fig. 5, which indicates a similar effect of changing the linear laser energy and the number of pulses.
- Figure 6: it is very strange the result for PU at 160 mm/s which shows a reduction of the processing efficiency, and is not discussed by the authors.
The reduction in processing efficiency for PU observed in Figure 6 is due to two factors - the diffraction of the laser beam at the edges of the hole with a high slenderness ratio and the interaction of the removed material particles with successive pulses incident on the surface of the processed material. A similar phenomenon would occur for the MVQ material with the use of higher processing parameters impossible to achieve at the available position.
- In general, tests should be performed at different replications, and data should be provided with the corresponding standard deviations.
The standard deviation of the estimation error of the geometrical size of an indentations was the subject of the authors' previous research [11, 12]. This issue was also investigated by the Hirox company, the author of the measurement method. The errors in measuring the volume of a indentations can reach considerable values for slender and deep indentations, and sometimes measurement is impossible. For indentations with the values we deal with in the article, the standard deviation does not exceed 20% of the measured value and does not affect the qualitative assessment of the results.
- Row 207-208 same scanning speed? Which speed? And which frequency?
For these energy ranges a frequency of 400 kHz and a scan speed of 160 mm / s were used.
- Row 214-216. The problem here is that the optimal chosen parameters correspond to the limits of the operating conditions. For example, at 400 kHz a scanning speed of 160 mm/s is suggested. However, it is not clear what happens if the scanning speed is increased. Analogously, an energy of 12 mJ is suggested, but what happens if the energy is increased? According to the results of figure 6 the processing efficiency is expected to increase.
The device at our disposal generates a maximum energy of 12.5 µJ and a maximum pulse frequency of 400kHz and we are not able to perform tests above this value. It should be assumed that the efficiency of processing with increasing these parameters increases - this applies in particular to MVQ material. The possible increase in machining efficiency depends on the slenderness ratio of the pit.
- Figure 8 a) is not clear, in particular it is not evident the position of microindentation. The overall dimensions of the sealing ring should be given.
The drawing has been changed.
- Figure 10 is not necessary, probably it would be better to report a scheme of the test better highlighting the piston and seal
The drawing has been deleted.
- Figure 11 shows a reduction of the friction force by the use of indented seals. However, it is not clear what would be the effect on the durability of the seals. The presence of indentations on the surface of the seal is expected to reduce the service life, do to easier material ablation during operation. This issue should be better addressed and the potential advantages of the developed approach should be compared with potential issues.
The research on the influence of the texture of the working surface of the mechanical seals shows that the texture does not deteriorate the durability of the seal. The rationale is as follows;
- Texture elements create reservoirs of the lubricant and provide better lubrication,
- debris and wear particles can accumulate in the texture cavities and are thus eliminated from the friction zone,
- texture depressions under favorable circumstances are the source of lubrication-improving hydrodynamic effects.
Test results for the durability of mechanical seals with a textured working surface show increased durability. The authors believe that the same may be the case with elastomeric seals. Detailed identification of this problem is planned in further research.

Reviewer 2 Report
The authors present a UV ps laser processing in the nanofabrication of elastomers. The author optimized the processing parameters for materials such as polyurethane and silicone. They also demonstrated a potential application of the proposed technique. This study is very interesting and the results are well discussed in the manuscript. However, the following problems need to be considered:
- In Fig. 2, there are some finer ring structures in the pattern. It is very interesting. The formations mechanism should be discussed.
- The interface effect plays an important role in the ablation process. Is there any interface effects in the experiment? The thickness of the materials (PU and MVQ) should be provided. Is it necessary to consider the heat conduction?
- As shown in Fig. 1, the overlapping of single pulses is a key point which should be discussed. Such as the influence of the lateral displacement.
- In fig. 2, there are two curves for the relationship between the 10 μJ laser energy and the removed volume of PU. It seems one of them should be removed.
- In the Application section, some fabrication parameters should be provided in detail. For example, the parameter for fabricating the sample in Fig. 9.
- The image resolution of Fig. 1 is too low.
- Some related reports should be cited, such as laser ablation in Polymers 12(4), 959 (2020), Adv. Mater. 23, 1860 (2011) and Opt. Express 23, 1863 (2015), melting method in Nanotechnology 25, 265302 (2014).
- Some typos should be corrected, such as an unnecessary letter in line 29, the legend in fig. 7.
Author Response
Responses to the review.
- In Fig. 2, there are some finer ring structures in the pattern. It is very interesting. The formations mechanism should be discussed.
Figure 2 shows the changes in the view of the microindentations with changes in frequency and pulse energy. Visible here are microindentations with various dominant ablation mechanisms.
- The interface effect plays an important role in the ablation process. Is there any interface effects in the experiment? The thickness of the materials (PU and MVQ) should be provided. Is it necessary to consider the heat conduction?
The thickness of the MVQ and PU samples was the same and amounted to 5 mm. In ultra-short pulse lasers, the influence of the thermal effect is minimal and can be neglected for the sample thickness used.
- As shown in Fig. 1, the overlapping of single pulses is a key point which should be discussed. Such as the influence of the lateral displacement.
During processing, we deal with the overlapping of individual pulses with a Gaussian energy density distribution for one beam trajectory (a single cavity). The mutual overlapping of the pulses depends on the operating parameters of the laser device and increases with the increase of the pulse energy. The efficiency of the process also depends on the diffraction of radiation on the walls of the cavity, as well as the interaction of radiation pulses with the material particles detached from the treated surface.
- In fig. 2, there are two curves for the relationship between the 10 μJ laser energy and the removed volume of PU. It seems one of them should be removed.
Fig. 6 correctly shows the operating parameters of the laser device (e.g. for MVQ and 10uJ energy there are two frequencies: 200 and 400kHz).
- In the Application section, some fabrication parameters should be provided in detail. For example, the parameter for fabricating the sample in Fig. 9.
The energy was 12.5 µJ and the scanning speed was 160 mm/s.
- The image resolution of Fig. 1 is too low.
The drawing was corrected.
- Some related reports should be cited, such as laser ablation in Polymers 12(4), 959 (2020), Adv. Mater. 23, 1860 (2011) and Opt. Express 23, 1863 (2015), melting method in Nanotechnology 25, 265302 (2014).
The citation has been completed.
- Some typos should be corrected, such as an unnecessary letter in line 29, the legend in fig. 7.
Typos have been corrected.

Reviewer 3 Report
The letter "c" at the end of the first paragraph should not be there.
Please introduce a reference for equations 1 and 2.
The paragraph "Theoretical basis" is more like a literature review and should be merged with the introduction.
The letters in figure 1 are not visible. They should be magnified. Specify the meaning of P, v and f when they first appear.
In figure 4 there is no indication of which figures are a), b) etc. Also the scales are not clearly visible.
What is the meaning of the solid lines in figures 5 and 7?
Author Response
Responses to the review.
- The letter "c" at the end of the first paragraph should not be there.
Typos have been corrected.
- Please introduce a reference for equations 1 and 2.
The source of the equation has been completed.
- The paragraph "Theoretical basis" is more like a literature review and should be merged with the introduction.
Chapter 2 is indeed based on the literature, but the authors believe that it should remain as a separate chapter.
- The letters in figure 1 are not visible. They should be magnified. Specify the meaning of P, v and f when they first appear.
The drawing has been changed. The listed values have been explained.
- In figure 4 there is no indication of which figures are a), b) etc. Also the scales are not clearly visible.
The drawing has been changed.
- What is the meaning of the solid lines in figures 5 and 7?
This is an approximation of the obtained results for the parameters used.
